# Large Language Models Are Stronger Entropy Models for Transform Coding

## Abstract

Large language models (LLMs) have shown promising advancements in lossless compression due to their excellent next-token prediction capabilities. However, there is a gap between LLM-based compressors and classical transform-based codecs. Existing LLM-based compressors function solely as entropy coders, focusing on compressing redundant data in the raw domain. In contrast, classical codecs typically transform raw data into more compact features in the latent domain before applying entropy coding. But LLM-based compressors have not discussed this case. To the best of our knowledge, this is the first work to introduce an LLM-based entropy model for transform coding. Specifically, we propose a simple yet effective fine-tuning strategy, tested across various codecs for both images and speeches. With less than 2% parameters are fine-tuned, the LLMs can serve as highly effective entropy models for well-established transform-based compression codecs. For instance, LLaMA3-8B paired with arithmetic coding compresses latent image codes on Kodak to 4.62% and speech codes on LibriTTS to 42.53% of their transformed sizes after fine-tuning. Our proposed methods achieve notable BD-rate improvements of 54.07% over JPEG, 17.61% over VQGAN, and 34.61% over SpeechTokenizer. These findings highlight the great potential of integrating LLMs into codecs to significantly improve coding efficiency. Source codes will be released upon acceptance.

## 1 Introduction

Large language models (LLMs) have achieved great success on various tasks and are highly-efficient for probabilistic prediction. According to source information theory (Shannon, 1948), probability prediction combined with entropy coding is equivalent to lossless data compression, as the minimum bit length required to represent data is determined by its $-\log_2$ probability likelihood (MacKay, 2003). Therefore, by combining LLM's predictive capabilities with entropy coding methods like Huffman coding (Huffman, 1952) and arithmetic coding (Pasco, 1977; Rissanen, 1976), LLMs can function as highly effective lossless compressors. Some works have already validated their competitive compression performance by feeding the text, images and audio data directly into LLMs, such as Valmeekam et al. (2023) and Deletang et al. (2024).

However, there is a clear gap between LLM-based compressors and classical transform-based codecs. Existing LLM-based compressors (Deletang et al., 2024; Valmeekam et al., 2023) solely play a role of entropy coders and process redundant data in its raw domain, while classical codecs (Wallace, 1992; Esser et al., 2021; Cheng et al., 2020; Liu et al., 2023b; Zhang et al., 2024b; Du et al., 2023) typically transform the data into latent feature domain, achieving more compact representations and then apply entropy coding to further improve compression efficiency. Hence in this paper, we bridge the gap by integrating LLM-based entropy models with transform coding frameworks. We first transform the data into compact latent domain and perform quantization. The quantized discrete codes are then fed into LLMs for context probability prediction, followed by entropy coding to generate the binary bitstream, see Fig. 1. Leveraging the powerful prediction capabilities of LLMs, we can further enhance the compression performance of classical transform-based codecs.

Although quantized latent codes can be treated as tokens for LLMs' input, two key challenges still remain. First, the latent codes are typically two- or multi-dimensional, which is incompatible with the sequential one-dimensional input format required by LLMs. A reasonable flatten method needs

Figure 1: **Left:** previous LLM-based compression methods (Deletang et al., 2024). **Right:** our proposed transform-based codec that replaces the entropy model with LLMs. T and IT represent transform and inverse transform, Q and IQ denote quantization and inverse quantization, and AE and AD stand for arithmetic encoding and decoding, respectively.

to be considered to preserve the correlation for the latent features before feeding them into LLMs. Second, the codes' value ranges do not align with LLMs' vocabulary size, leading to inconsistency in LLMs' probability predictions, as LLMs can only predict within their predefined vocabulary range. Thus, we introduce a latent-codes arrangement module between quantization and LLM-based entropy model. This module maps the quantized latent codes to a positive range and determines the vocabulary size based on the value range. We only adjust the dimensions of LLMs' input and output layers to match the specified vocabulary size, then freeze the LLMs' backbone and fine-tune the two layers to better adapt to the characteristics of various codecs.

In this paper, we demonstrate that large language models are stronger entropy models for transform-based codecs. We integrate LLMs into transform coding frameworks and apply simple fine-tuning to adapt them for compression across various modalities. Experiments are conducted on three types of codecs to validate our methods' effectiveness: codecs using the classic discrete cosine transform (DCT) and quantization like JPEG (Wallace, 1992), codecs using the neural network-based transform and vector quantization (VQ) like VQGAN (Esser et al., 2021), and codecs using network-based transform and residual vector quantization (RVQ), such as SpeechTokenizer (Zhang et al., 2024b). Our method demonstrates satisfactory performance across all these codecs, notably improving their compression efficiency.

Generally speaking, our contributions can be summarized as

- We extensively evaluate recently-released LLMs in lossless text compression and select Llama3-8B (AI@Meta, 2024) as the backbone for our proposed entropy model.
- We propose a latent-codes arrangement module to align the dimension between LLMs and latent codes, along with a simple yet effective fine-tuning strategy to adapt LLM-based entropy models to various compressors.
- Experiments show that our LLM-based entropy model can significantly improve the compression efficiency of image codecs like JPEG and VQGAN by 54.07% and 17.61%, and enhance speech codecs like SpeechTokenizer by 34.61%.

## 2 RELATED WORKS

**Classical Codecs** Classical image and speech compression techniques have long relied on transform-based coding frameworks. Image codecs like JPEG and BPG (Bellard, 2014) typically use DCT to transform the raw data into latent domain, while speech codecs like Opus (Valin et al., 2012), AMR-WB (Sjoberg et al., 2007), and AAC (iso, 2006) employ transforms like MDCT and STFT. All these codecs reduce latent data redundancy through entropy coding methods like Huffman or arithmetic coding. However, instead of dynamically estimating data probabilities, they rely on pre-calculated frequency tables based on empirical data, which limits their compression efficiency.

**Neural Codecs** In recent years, modern deep learning methods like variational autoencoders (VAE) (Kingma & Welling, 2022) and neural network-based codecs (Cheng et al., 2020; Liu et al., 2023a; Zeghidour et al., 2021; Défossez et al., 2022; Du et al., 2023) have demonstrated promising

Table 1: Compression performance of different LLMs on enwik9. Compression ratio represents the ratio of compressed size to raw data size. Chinchilla is a closed-sourced, thus the data at * cannot be measured.

| Chunk Size | Compression Ratio (%)↓ | | | | | | | | |
|---|---|---|---|---|---|---|---|---|---|
| | Chinchilla | | | RWKV-v6 | | ChatGLM3-6B | GLM4-9B | Llama2-7B | Llama3-8B |
| | 1B | 7B | 70B | 1B6 | 3B | | | | |
| 1024 | * | * | * | 10.80 | 10.59 | 18.03 | 9.37 | 9.81 | **8.96** |
| 2048 | 11.3 | 10.2 | 8.3 | 10.54 | 9.86 | 16.27 | 8.83 | 8.87 | **7.81** |
| 4096 | * | * | * | 10.08 | 9.4 | 14.67 | 8.07 | 8.30 | **7.34** |

performance in both speech and image compression. These methods utilize elaborate neural networks to transform raw data into latent domain, and employ sophisticated entropy models to predict latent features' distributions, thereby improving the compression efficiency. However, their entropy models often require complex modality-specific design of network architectures, which poses challenges for developing a unified compression framework for multiple modalities.

**LLM-based Compressors**    LLMs have been explored as effective tools for lossless data compression due to their strong probabilistic prediction capabilities. Valmeekam et al. (2023) directly leverages LLMs' text processing capability and proposes an efficient text compressor based on LLMs. Deletang et al. (2024) further demonstrates that LLMs can serve as universal compressors on multimodal data like text, image, and audio. However, these methods directly compress data in the redundant raw data domain. In this paper, we further investigate the potential of LLMs as strong entropy models within transform-based coding frameworks.

## 3   BACKGROUND AND BENCHMARK

### 3.1   BACKGROUND

Given a sequence of data $x_{1:n}$ with distribution $p_{1:n}$, lossless compression methods aim to encode the data into binary streams $C(x_{1:n})$, minimizing its bit length $l_c(x_{1:n})$ while preserving the original information. According to the source-coding theorem, the expected minimal $l_c(x_{1:n})$ is given by its entropy $E_{x \sim p}[-\log_2 p]$. In existing LLM-based compressors, $x$ represents words, pixels, and speech samples for text, image, and speech compression, respectively.

However, the actual data distribution is usually unknown during compression. Therefore, we autoregressively predict the distribution $\hat{p}_{1:n}$ using LLMs during encoding and decoding, where the probability space of is confined to the LLMs' vocabulary size, requiring that both the input and output layers of LLMs must match this dimensionality for accurate prediction. Upon applying arithmetic coding, the generated bit length can be approximated via cross-entropy as

$$H(x) = E_{x \sim \hat{p}} \left[ \sum_{i=1}^{n} -\log_2 \hat{p}\left(x_i \mid x_{1:i-1}\right) \right] \tag{1}$$

### 3.2   BENCHMARK

To select a suitable LLM as the backbone of the entropy model, we evaluate the compression capabilities of several recent LLMs on lossless text compression. Evaluations are conducted using the pre-trained models and their corresponding tokenizers on the enwik9 dataset (Hutter, 2006). Involved open-source LLMs include LLaMA2-7B (Touvron et al., 2023), LLaMA3-8B (AI@Meta, 2024), RWKV-v6 (Peng et al., 2024), ChatGLM3-6B (Wang et al., 2023), and GLM4-9B (GLM et al., 2024). For comprehensive comparison, we also assess the closed-source Chinchilla model (Hoffmann et al., 2022), using its compression performance reported in Deletang et al. (2024).

Considering LLMs' context limitations, the text tokens are divided into chunks before feeding into LLMs for compression. Table. 1 represents the compression performance under various chunk sizes. While all these LLMs demonstrate pleasing lossless text compression capabilities, Llama3-8B stands out with the best compression efficiency and longer context supports (up to 8192), making it

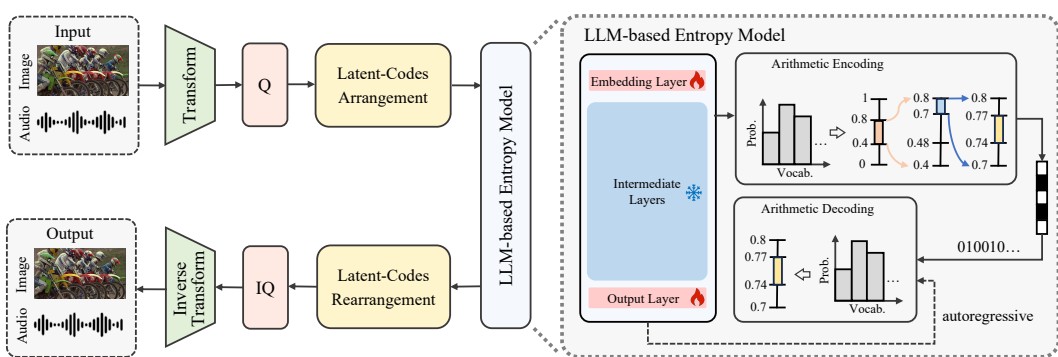

Figure 2: The proposed transform-based coding framework uses LLMs as the entropy model. Raw data is transformed and quantized into compact latent codes, which are flattened and fed into the LLM-based entropy model in chunks. The model, consisting of fine-tuned LLMs and arithmetic coding, compresses the data into a bitstream. On the decoding side, the process is reversed to reconstruct the data.

the optimal backbone for our experiments. By contrast, other LLMs, limited to a maximum context length of 4096, achieve relatively longer compression time and reduced compression efficiency.

## 4 PROPOSED METHOD

This section presents our LLM-based transform coding framework, demonstrating its effectiveness across three types of codecs: (1) codecs using the classic DCT and quantization approach, such as JPEG; (2) codecs using the neural network-based transform and vector quantization, like VQGAN; (3) codecs using network-based transform and residual vector quantization, like SpeechTokenizer.

### 4.1 OVERALL FRAMEWORK

**Workflow** Fig. 2 illustrates the proposed LLM-based transform coding framework. On the encoding side, raw data $x$ is transformed by $T(\cdot)$ into latent features $y$. Then, $Q(\cdot)$ quantizes these features into discrete latent codes $\hat{y}$ with a distribution $p$. The process is described as:

$$\hat{y} = Q(T(x)) \tag{2}$$

The latent codes are discrete integer values and can be directly treated as tokens for LLMs' input. However, the latent codes are typically multi-dimensional and not compatible with the one-dimensional input format required by LLMs. Hence we first flatten the codes in raster order to preserve spatial correlations. Due to LLMs' context limitations, we split the flattened codes into several chunks $\hat{y}_{1:n}$ of size $n$. Each chunk is fed into the LLMs sequentially, where the conditional probability $\hat{p}(\hat{y}_i \mid \hat{y}_{1:i-1})$ is calculated based on its context. Arithmetic coding is then applied to compress the latent codes to their entropy limit and optimize coding efficiency. The expected bit length is determined by cross-entropy like eq. (1) and can be formulated as eq. (3).

$$H(\hat{y}) = E_{\hat{y}\sim\hat{p}} \left[ \sum_{i=1}^{n} - \log_2 \hat{p}\left(\hat{y}_i \mid \hat{y}_{1:i-1}\right) \right] \tag{3}$$

On the other hand, the value ranges of latent codes do not align with the pre-trained LLMs' original vocabulary size, which causes inconsistencies in the LLMs' probability predictions, as they are restricted to the predefined vocabulary. Therefore, we introduce a latent-codes arrangement module between quantization and LLM-based entropy model. This module maps the quantized latent codes into positive ranges and defines the vocabulary size accordingly. We then adjust the LLMs' input and output dimensions to match the vocabulary size, freezing the LLMs' backbone and fine-tuning only the input and output layers to adapt it to different codecs.

On the decoder side, the latent codes $\hat{y}$ are autoregressively decoded using the same fine-tuned LLMs and an arithmetic decoder. After being reversed to the original formats, it is reconstructed to

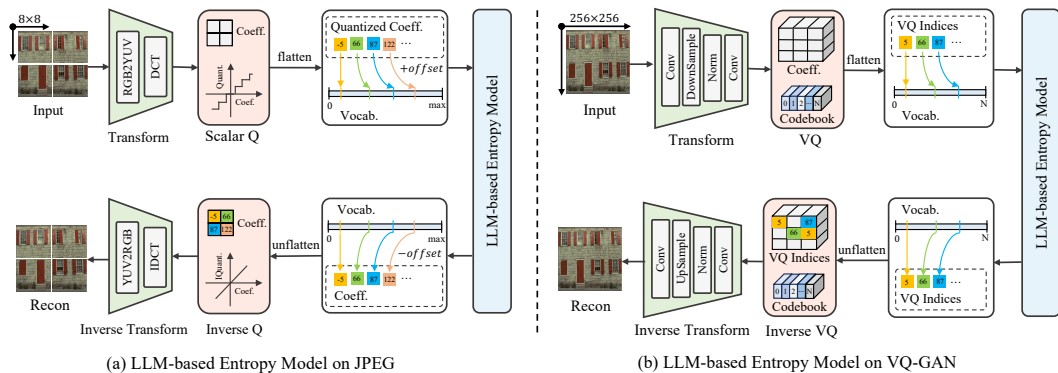

(a) LLM-based Entropy Model on JPEG          (b) LLM-based Entropy Model on VQ-GAN

Figure 3: Proposed LLM-based entropy model on JPEG (**Left**) and VQGAN (**Right**).

$\hat{x}$ through inverse quantization $IQ(\cdot)$ and inverse transform $IT(\cdot)$ as:

$$\hat{x} = IT(IQ(\hat{y})) \tag{4}$$

**Latent-Codes Arrangement** To align the LLMs' input and output dimension (i.e., the vocabulary size) with the latent codes' value range, we design a latent-codes arrangement module between quantization and proposed LLM-based entropy module. JPEG's latent codes include both positive and negative values, hence we add an offset to ensure all values are non-negative and generate the LLMs' vocabulary. For VQGAN, which uses VQ and has a predefined codebook, we directly use the codebook as LLMs' vocabulary. SpeechTokenizer employs RVQ with layers having distinct codebooks, we assign a specific vocabulary for each layer.

**Fine-tuning of LLMs** Upon determining the vocabulary size based on latent codes' value ranges, we modify the LLMs' input and output dimensions accordingly. To better adapt the modified LLMs to each codec, we then freeze its backbone and fine-tune only the input and output layers. During fine-tuning, the prediction likelihood is maximized by minimizing the cross-entropy loss, which aligns with the goal of compression. Experiments demonstrate that with less than 1% parameters updated, the fine-tuned LLMs can effectively capture the latent codes' characteristics and achieve satisfactory performance across various datasets.

## 4.2 IMAGE COMPRESSION WITH LLM-BASED ENTROPY MODEL

For image compression, we utilize the highly representative traditional codec JPEG and the widely adopted image generation method VQGAN as anchor codecs, replacing their entropy models with our proposed LLM-based entropy model, as shown in Fig. 3.

**JPEG** As shown in Fig. 3 (a), with JPEG as the anchor codec, raw images are divided into $8 \times 8$ patches and transformed into DCT domain. After scalar quantization, the latent codes are flattened in raster order and an offset is added to ensure all values are non-negative before being fed into the LLMs. Though JPEG's latent codes do not have a fixed maximum value, tests on several datasets reveal that most values fall between -127 and 128. Hence we empirically set the offset to 127 and the maximum value to 255, with all exceeding values truncated to 255, which means that the generated vocabulary size is 256. By altering the quality factors, variable bitrate compression can be achieved.

**VQGAN** When using VQGAN as the anchor codec, raw images are transformed into the latent domain via a neural network without being divided into patches, as shown in Fig. 3 (b). Then, they are mapped to discrete integer values using a fixed codebook through vector quantization. Therefore, the LLMs' vocabulary size is directly determined by VQGAN's codebook. VQGAN supports several architectures with different codebook sizes and corresponding different bitrates, hence we also conduct experiments to compare the compression performance using different codebook sizes. During fine-tuning, $256 \times 256$ input images are quantized into latent codes of size $16 \times 16$ and flattened to length of 256 before being fed into LLMs. During inference, latent codes of different

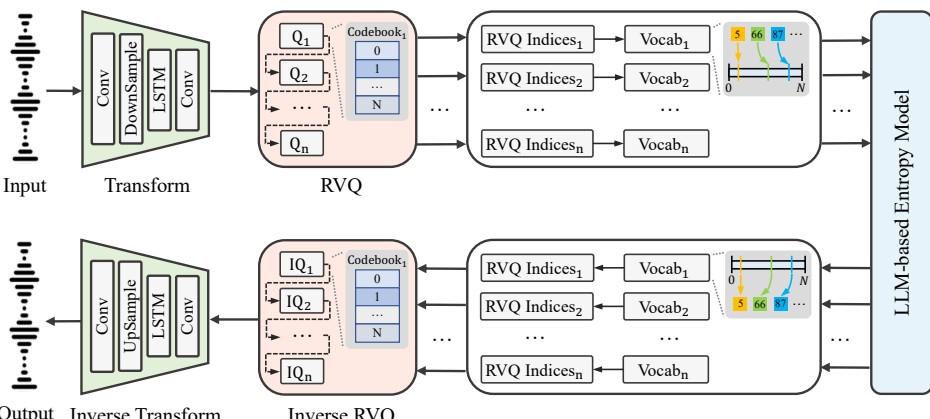

Figure 4: Framework of LLM-based entropy model on SpeechTokenizer. Layered RVQ indices are treated as latent codes, quantized from latent features by several independent RVQs. RVQ indices of each layer make latent-codes arrangements respectively.

images can be concatenated to fully leverage the LLMs' contextual capability, enabling simultaneous compression of multiple images.

## 4.3 Speech Coding with LLM-based Entropy Model

When using speech codecs like SpeechTokenizer as anchor codecs, the raw speech is downsampled and transformed by neural networks and then quantized hierarchically using residual vector quantization. The quantization layers share the same codebook but are mutually independent. Hence we perform a separate latent-codes arrangement for each layer, as shown in Fig. 4.

Speech of length $L$ is quantized into 8 equal-sized latent codes using eight-layer RVQ, with each code containing $N/320$ values. We train separate LLMs for each layer, processing latent codes sequentially. Typically, a single quantization layer produces fewer than 1024 latent codes, which is situated within almost all LLMs' context length. Hence it is unnecessary to chunk the input speeches. During decoding, speeches are reconstructed by concatenating RVQ layers from the first to the $n$-th. Changing the number of layers allows for variable speech quality, with the bitrate adjusted accordingly. During inference, multiple speech inputs can be concatenated to maximize the GPU utilization and enable simultaneous compression.

## 5 Experiments

### 5.1 Experimental Setup

**Implementation Details**  During LLMs' training process, we freeze the LLMs' backbone and fine-tune only the input and output layers. We use AdamW optimizer (Loshchilov & Hutter, 2019) with an initial learning rate of $2.5 \times 10^{-4}$ and a cosine annealing scheduler with warm restarts. The $\beta_1$ and $\beta_2$ values are set to 0.9 and 0.99, respectively, with a weight decay of $1 \times 10^{-2}$ to prevent overfitting. Automatic mixed precision training is employed to reduce GPU memory usage and accelerate both training and inference. Fine-tuning runs for 5 epochs per dataset, with quantized latent codes pre-extracted to further save GPU memory. All experiments are conducted on an NVIDIA GTX 4090 GPU with 24GB of memory.

**Datasets**  We fine-tune the LLMs for image compression using a subset of ImageNet database (Russakovsky et al., 2015) and cropped them into 13,830 samples with the size of 256 $\times$ 256. We evaluate performance on a validation subset of ImageNet, Kodak (Kodak, 1999) and CLIC (CLIC, 2021) datasets. For speech compression, LLMs are fine-tuned on the LibriTTS training dataset (Zen et al., 2019), with compression performance assessed using the LibriTTS clean-test and other-test dataset, high-quality LJSpeech (Ito & Johnson, 2017) dataset.

Table 2: Compression performance of LLM-based entropy models in various codecs and datasets. "QF" and "CB" are JPEG quality factor and VQGAN codebook size, respectively. "RVQ-1:2" denotes the sum of the first two layers. While "CR" is the compression ratio of latent codes, "BD-Rate" reflects the overall compression efficiency of raw data. JPEG and VQGAN use chunk sizes of 2048 and 256, respectively. More details can be seen in appendix A.1.

| Anchor Codecs | Settings | Dataset | CR(%)↓ | BD-rate(%)↓ | Avg.(%) |
|---|---|---|---|---|---|
| JPEG | QF-20 | Imagenet-val | 4.43 | -53.05 | -54.07 |
| | QF-50 | | 6.98 | | |
| | QF-80 | | 8.79 | | |
| | QF-20 | Kodak | 4.62 | -53.49 | |
| | QF-50 | | 7.58 | | |
| | QF-80 | | 9.36 | | |
| | QF-20 | CLIC | 4.96 | -55.66 | |
| | QF-50 | | 7.45 | | |
| | QF-80 | | 9.19 | | |
| VQGAN | CB-1024 | Imagenet-val | 90.56 | -17.42 | -17.61 |
| | CB-16384 | | 75.31 | | |
| | CB-1024 | Kodak | 90.70 | -17.30 | |
| | CB-16384 | | 75.41 | | |
| | CB-1024 | CLIC | 89.78 | -18.11 | |
| | CB-16384 | | 74.70 | | |
| SpeechTokenizer | RVQ-1 | LibriTTS-clean | 42.53 | -35.35 | -34.61 |
| | RVQ-1:2 | | 67.36 | | |
| | RVQ-1:8 | | 89.59 | | |
| | RVQ-1 | LibriTTS-other | 49.57 | -31.99 | |
| | RVQ-1:2 | | 70.80 | | |
| | RVQ-1:8 | | 91.20 | | |
| | RVQ-1 | LJSpeech | 42.17 | -36.49 | |
| | RVQ-1:2 | | 66.59 | | |
| | RVQ-1:8 | | 89.39 | | |

**Metrics** We evaluate image compression performance using Fréchet Inception Distance (FID) (Heusel et al., 2017) and Peak Signal-to-Noise Ratio (PSNR) distortion metrics. FID evaluates the distributional distance between the reconstructed and original images, while PSNR measures the pixel-level similarity. For speech compression, we use the Virtual Speech Quality Objective Listener (VISQOL) (Hines et al., 2012) to assess perceptual quality and the Word Error Rate (WER) (Ali & Renals, 2018) to measure transcription errors.

## 5.2 COMPRESSION PERFORMANCE

Table. 2 provides the compression performance of LLM-based entropy models across various anchor codecs and datasets. Three quality factors (20, 50, 80) and two codebook sizes (1024, 16384) are evaluated for JPEG and VQGAN, respectively. While "RVQ-1" represents the first RVQ layer's compression performance, "RVQ-1:2" denotes the cumulative performance of the first two layers. "CR" is the lossless compression ratio of latent codes purely caused by the LLM-based entropy model, "BD-Rate (Bjontegaard, 2001; Pateux & Jung, 2007)" measures the overall compression efficiency of raw data. Herein, JPEG and VQGAN are evaluated using chunk sizes of 2048 and 256, respectively. The chunk size of SpeechTokenizer is set to 512.

For each anchor codec, we evaluate the performance on three datasets to prove its robustness. For DCT-based codecs like JPEG, the latent codes are highly sparse. Hence our LLM-based entropy model can compress them to less than 10% of the original size, achieving notable BD-Rate improvements of 54.07%. By contrast, neural network-based codecs like VQGAN exhibit lower redundancy

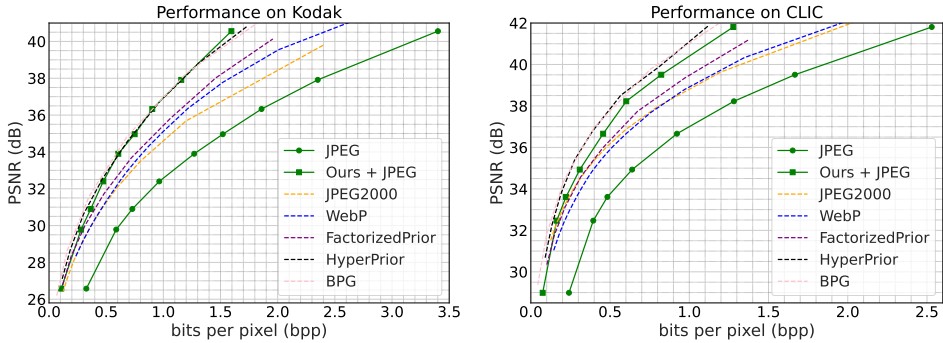

Figure 5: Compression performance on Kodak (**Left**) & CLIC (**Right**) dataset, using JPEG as an anchor codec.

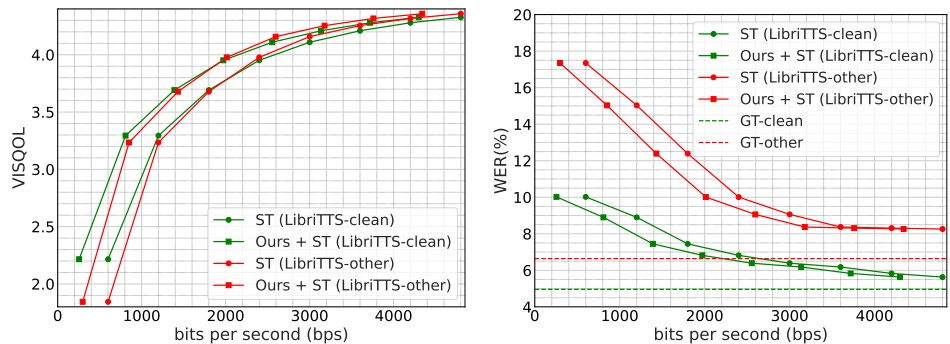

Figure 6: Compression performance of VISQOL(**Left**) and WER(**Right**) on speech datasets, using SpeechTokenizer (ST) as an anchor codec. GT represents the ground truth, calculated from the original speech.

in the latent space, resulting in limited compression ratios (about 70%) of latent size and moderate improvements of BD-Rate (17.61% on average). In SpeechTokenizer, which uses RVQ for quantization, the first RVQ layer contains richer semantic information, leading to better compression performance. As more layers are added, the compression efficiency declines. Varying layer depths lead to different bit rates, with an overall BD-Rate gain of 34.61%.

The overall RD curves for JPEG codecs are illustrated in Fig. 5, the proposed LLM-based entropy model significantly enhances JPEG's compression performance, outperforming more advanced codecs like JPEG2000 (Christopoulos et al., 2000), WebP (Si & Shen, 2016), and FactorizedPrior structure in Ballé et al. (2018), even nearing the efficiency of BPG and HyperPrior structure in Ballé et al. (2018). At the same bitrate, our enhanced JPEG codec achieves approximately 4dB improvements over the original JPEG on both Kodak and CLIC datasets. In Fig. 6, it can be seen that our entropy model improves the speech compression, as reflected by gains in VISQOL and WER metrics, where the best efficiency is achieved in only compressing the first RVQ layer.

Table 3: Percentage of fine-tuned parameters for different vocabulary sizes in Llama3-8B.

| Vocabulary Size | 256 | 1024 | 2048 | 4096 | 8192 | 16384 |
|---|---|---|---|---|---|---|
| Fine-tuned Param.(M) | 2.1 | 8.4 | 16.8 | 33.6 | 67.1 | 134.2 |
| Percentage(%) | 0.026 | 0.105 | 0.210 | 0.419 | 0.839 | 1.678 |

### 5.3 DISCUSSIONS

**Number of Finetuned Parameters**    To adapt the LLMs to various codecs, we modify their input and output dimensions to newly defined vocabulary size and fine-tune these two layers. Table. 3 illustrates the percentage of parameters that requires fine-tuning for different vocabulary sizes. It can be seen that the proportion of fine-tuned parameters remains below 2% even for a large vocabulary

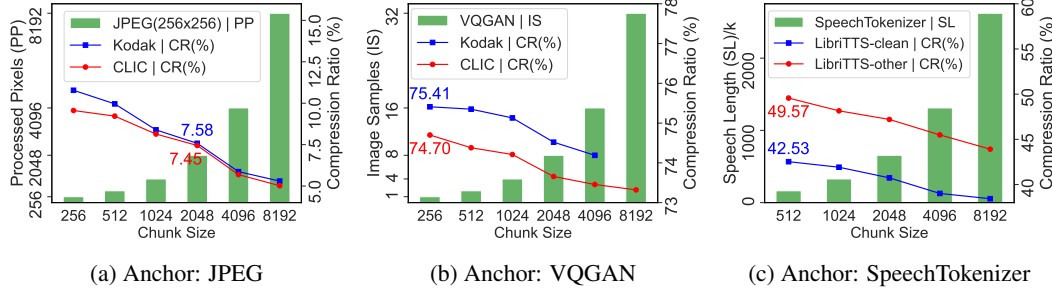

|  |  |  |
|---|---|---|
| (a) Anchor: JPEG | (b) Anchor: VQGAN | (c) Anchor: SpeechTokenizer |

Figure 7: Effect of chunk size on different anchor codecs. "PP" in (a) refers to processed pixels in a single image chunk, "IS" in (b) denotes the number of image samples, while "SL" in (c) is the speech length.

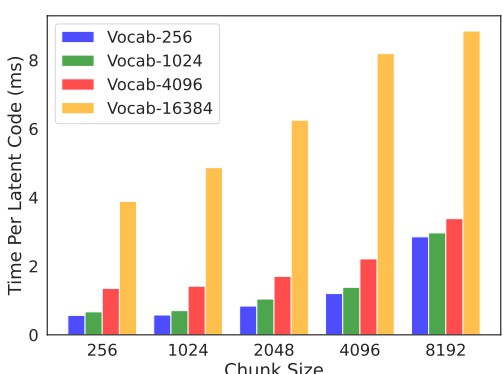

Figure 8: Inference time (ms) per latent code with different vocabulary sizes.

Table 4: Running Time (s) of different anchor codecs. The size of processing image is $64 \times 64$ for JPEG, and $256 \times 256$ for VQGAN. Total running time is calcuated by time per codes $\times$ chunk size $\times$ chunk number.

| Chunk | Running Time (s) | | |
|---|---|---|---|
| Size | JPEG | VQGAN | VQGAN |
|  | Vocab-256 | Vocab-1024 | Vocab-16384 |
| 256 | 2.33 | 0.171 | 0.99 |
| 512 | 2.34 | 0.174 | 1.05 |
| 1024 | 2.37 | 0.180 | 1.25 |
| 2048 | 3.43 | 0.267 | 1.60 |
| 4096 | 4.92 | 0.354 | 2.10 |
| 8192 | 11.69 | 0.759 | 2.26 |

size of 16,384. Though only a small subset of parameters being fine-tuned, the updated LLMs still exhibit well generalization across various codecs.

**Effect of Chunk Size on Compression Ratio** According to the latent size, our anchor codecs fall into two categories. The first, like JPEG, generates latent codes matching the original image size, requiring images to be split into multiple chunks due to LLMs' context limitation. The second type, such as VQGAN and SpeechTokenizer, transforms raw data into dowm-sampled latent codes, allowing multiple tokens to be combined and compressed in a single input.

Fig. 7 illustrates the compression performance across chunk sizes from 256 to 8192. The boxplots "PP", "IS", and "SL" in (a), (b), and (c) represent processed pixels per image chunk for JPEG, the number of image samples in VQGAN, and speech length in SpeechTokenizer, respectively. Larger chunk sizes allow LLMs to utilize more contextual information, significantly improving the compression ratio of latent codes (shown on the right y-axis). For JPEG, the latent's compression ratio improves from 10.78% to 5.3% on Kodak and 9.56% to 5.02% on CLIC. For VQGAN and Speech-Tokenizer, whose latent compression ratio are relatively higher, increasing the chunk size boosts the compression ratio by about 1% and 5%, respectively.

**Limitations in Compression Complexity** Our LLM-based entropy model requires autoregressive probability predictions during both encoding and decoding. Fig. 8 provides the inference time per latent code for different vocabulary sizes and chunk sizes. As the vocabulary size increases, the probability space for each latent code expands. resulting in significant increases in inference time. Similarly, when the chunk size grows, each code's prediction context becomes longer, which also slows down the inference time. To ensure an acceptable encoding and decoding time, we typically use chunk sizes less than 2048. Recent developments in lightweight LLMs (Abdin et al., 2024; Team et al., 2024; Zhang et al., 2024a) also provide more options for our future work.

Table. 4 shows the running time required by our method to process a single image with different anchor codecs and chunk sizes. The running time here refers to the sum of entropy encoding and

decoding time. Without parallel computation, the complexity of our autoregressive method is proportional to the size of latent codes. Actually, recent neural codecs (Esser et al., 2021; Zhang et al., 2024b; Cheng et al., 2020; Liu et al., 2023b) generate latent codes with a downsampled size relative to the image size. By applying the LLM-based entropy model to latent codes instead of raw data, the running time of the autoregressive LLM-based entropy model is reduced to a more practical level.

**Effect of Flatten Order on Compression Ratio**   When using JPEG as the anchor codec, latent codes are flattened in zigzag order and compressed using Huffman coding. However, LLMs' inputs are typically flattened in raster scan order. We compare the compression ratios of latent codes flattened in either zigzag or raster order, using the proposed LLM-based entropy model. As shown

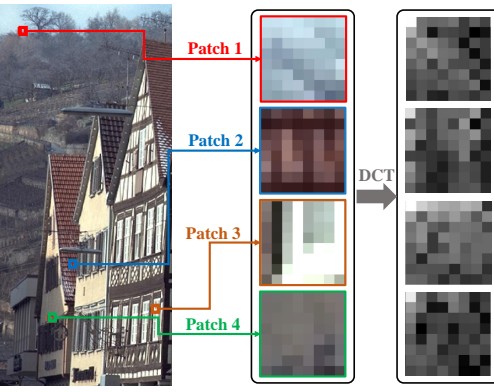

Figure 9: The visualization of $8 \times 8$ patches and their corresponding DCT spectra of luminance channel, extracted from the Kodak dataset.

in Table. 5, raster order flattening consistently yields more efficient compression across all datasets and quality factors, with the performance advantages becoming more pronounced at higher bitrates (about 0.38% to 0.49% at QF-20 on Kodak and CLIC). It is likely because raster order preserves the correlations of different frequencies (i.e., latent codes), enabling LLMs to learn a more accurate probability distribution and achieve better compression efficiency. For instance, it can be observed in Fig. 9 that DCT coefficients still have strong two-dimensional correlations in frequency domain. Specifically, each coefficient is correlated to its neighboring coefficients.

Table 5: Compression performance for latent codes flattened in Zigzag or Raster scan order on different image datasets, using JPEG as anchor codec. Given that the input image is $256 \times 256$. Chunk size is set to 2048. $\Delta$ represents the difference between compression ratio in zigzag order and that in raster order.

| | Compression Ratio (%)↓ | | | | | |
| | Kodak | | | CLIC | | |
| **Scan Order** | QF-20 | QF-50 | QF-80 | QF-20 | QF-50 | QF-80 |
|---|---|---|---|---|---|---|
| Zigzag | 5.11 | 7.79 | 9.44 | 5.34 | 7.68 | 9.33 |
| Raster | **4.62** | **7.58** | **9.36** | **4.96** | **7.45** | **9.19** |
| $\Delta$ | 0.49 | 0.21 | 0.08 | 0.38 | 0.23 | 0.14 |

**Future Works**   Though this research has achieved notable BD-rate improvements on several classical speech and image codecs, we will continue to evaluate LLM-based entropy models on more advanced SOTA codecs. We will enhance compression efficiency through improved training strategies, and explore low-complexity LLMs design to build efficient multimodal data compressors.

## 6   CONCLUSION

To the best of our knowledge, this is the first work to explore the potential of large language models as powerful entropy models in widely-used transform coding frameworks for images and speeches. We introduce a latent code arrangement module, and propose a simple yet effective fine-tuning strategy that adjusts less than 2% of the model parameters. We test our proposed method on various types of image and speech codecs, including JPEG with traditional DCTs and scalar quantization, VQGAN with neural transform and vector quantization, and SpeechTokenizer with neural transform and residual vector quantization. Experiments show that LLaMA3-8B, combined with arithmetic coding, reduces the latent size of image and speech codecs up to 4.62% on Kodak and 42.53% on LibriTTS, respectively. Consequently, our approach achieves significant BD-Rate improvements, outperforming classical codecs like JPEG, VQGAN, and SpeechTokenizer by 54.07%, 17.61%, and 36.61%, respectively.

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

## A    APPENDIX

### A.1    DETAILED EXPLANATION ON COMPRESSION RATIO AND BD-RATE

Here, we provide a more detailed explanation of the results in Table. 2 on three different types of codecs, JPEG, VQGAN, and SpeechTokenizer.

#### A.1.1    JPEG

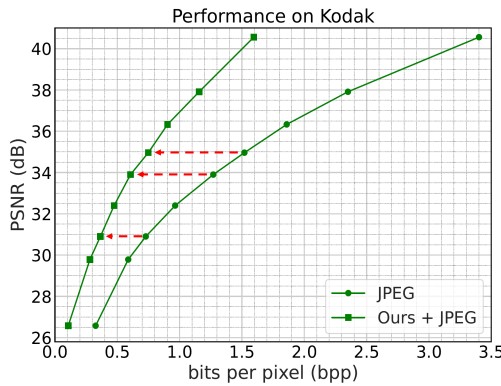

Figure 10: Compression performance on raw data using original JPEG and JPEG with our proposed LLM-based entropy model.

Table 6: Compression ratio on latent codes using Huffuman coding of original JPEG and our LLM-based entropy model, on Kodak dataset.

| Settings | Bits Per Pixel (bpp) | | |
|---|---|---|---|
| | QF-20 | QF-50 | QF-80 |
| Latent | 8 | 8 | 8 |
| JPEG (Huffman) | 0.72 | 1.27 | 1.52 |
| *CR(%)↓* | *8.94* | *15.88* | *19.01* |
| Ours (LLMs) | 0.370 | 0.606 | 0.749 |
| *CR(%)↓* | *4.62* | *7.58* | *9.36* |
| **BD-rate(%)↓** | | -53.49 | |

On JPEG, we test the compression ratio of the proposed LLM-based entropy model for latent codes and its improvement over the compression performance of JPEG on raw data, quantified by BD-rate. Specifically, as shown in Table. 6, the compression ratio here refers to the ratio of the compressed bitstream size to the quantized latent code size. By comparing the compression of latent codes using Huffman coding and our proposed LLM-based entropy model, we can observe the improvement in JPEG compression performance in raw data with our LLM-based entropy model, as shown in Fig. 10. Leveraging the two rate-distortion curves, we can quantify the improvement that our LLM-based entropy model brings to the compression performance of anchor codecs on raw data using the BD-rate.

By setting different quality factors, we can get the compression performance of the LLM-based entropy model at different bpps. We construct a vocabulary for the quantized coefficients by adding an offset of 128 and setting a maximum value of 255. In most image datasets, we find that the processed quantized coefficients generally do not exceed 255. For special cases where they do, we consider the loss accordingly. With a vocabulary size of 256, the bits per pixel of latent codes here is 8.

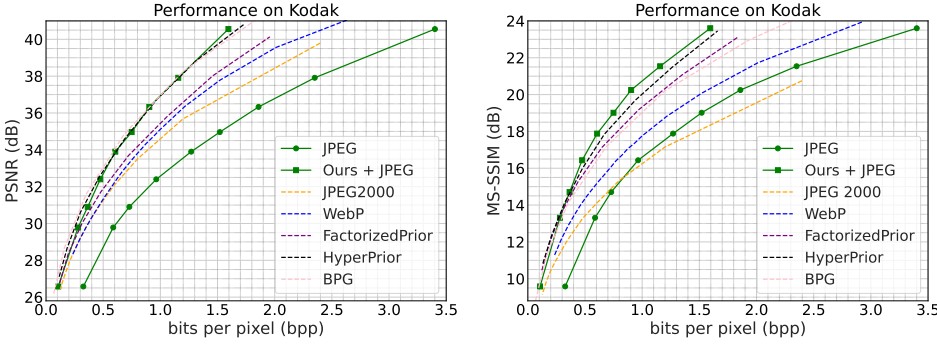

Figure 11: Compression performance on Kodak dataset in PSNR (**Left**) and MS-SSIM (**Right**), using JPEG as an anchor codec.

As shown in Fig. 11, in addition to PSNR, we also evaluate the compression performance of various methods on the Kodak dataset using MS-SSIM. For clearer comparison, here we converted MS-SSIM to $-10\log_{10}(1-\text{MS-SSIM})$.

### A.1.2 VQGAN

In VQGAN, we achieve further compression of the VQ indices, with varying compression ratios depending on the size of codebook, as shown in Table. 7 and Fig. 12. The bits per pixel of the VQ indices here can be calculated with eq. (5):

$$\text{bpp} = \frac{\text{Num. of Indices} \times \log_2(\text{Vocab. Size})}{\text{Num. of Pixels}} \tag{5}$$

Using VQs with different codebook sizes produces latent codes with varying quantization levels, enabling variable bitrates in LLM-based transform coding under the VQGAN anchor. The BD-rate here still reflects the improvement achieved by our proposed LLM-based entropy model in compressing raw data via VQGAN.

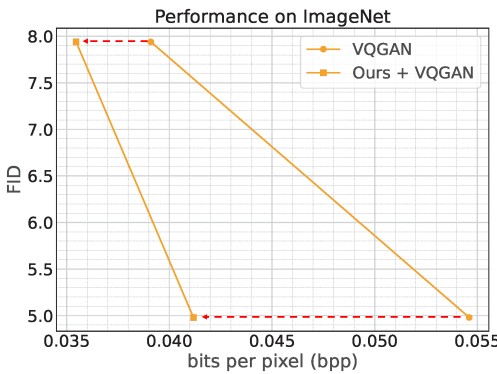

Figure 12: Compression performance on raw data using original VQGAN and VQGAN with our proposed LLM-based entropy model.

Table 7: Compression ratio on VQ indices by VQGAN anchor using our proposed LLM-based entropy model, on Imagenet-val dataset. "CB-1024" denotes the codebook size of VQGAN is 1024.

|  | Bits Per Pixel (bpp) | |
| --- | --- | --- |
| Settings | CB-1024 | CB-16384 |
| Latent | 0.0391 | 0.0546 |
| +Ours (LLM) | 0.0355 | 0.0412 |
| **CR(%)** | **90.56** | **75.31** |
| **BD-rate(%)** | **-17.42** | |

### A.1.3 SPEECHTOKENIZER

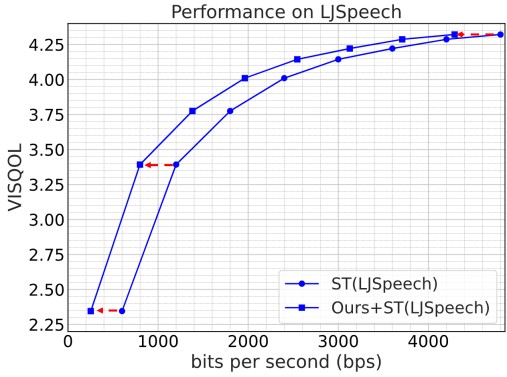

Figure 13: Compression performance on raw data using original SpeechTokenizer and SpeechTokenizer with our proposed LLM- based entropy model.

Table 8: Compression ratios on RVQ indices from different layers using LJSpeech dataset. "RVQ-1:2" denotes the sum of first and second layer.

|  | Bits Per Second (bps) | | |
| --- | --- | --- | --- |
| Settings | RVQ-1 | RVQ-1:2 | RVQ-1:8 |
| Latent | 600 | 1200 | 4800 |
| +Ours (LLM) | 253.02 | 799.08 | 4290.72 |
| *CR(%)*↓ | 42.17 | 66.59 | 89.39 |
| **BD-rate(%)↓** | | **-36.49** | |

On SpeechTokenizer, we further compress RVQ indices from different layers, as is shown in Table. 8. Compression ratio here stands for the ratio of compressed bitstream to VQ indices size. The indices from first layer of RVQ can be effectively compressed. Because the indices here contain only semantic information.

During decoding, by concatenating the RVQ indices from the first to the n-th layer, the raw speech can be reconstructed. As is shown in Fig. 13, by reducing the number of concatenated layers, speech with varying quality can be obtained. The bitrate used for reconstruction changes accordingly, achieving variable bitrate for LLM-based transform coding under SpeechTokenizer anchor. BD-rate shows the improvement of SpeechTokenizer in compressing raw data enhanced by our proposed LLM-based entropy model.

## A.2 COMPRESSION PERFORMANCE ON FREQCODEC (DU ET AL., 2023)

Apart from SpeechTokenizer, we also apply the proposed LLM-based entropy model to build speech transform coding on FreqCodec. FreqCodec uses network-based transforms and 32-layer RVQs, similar to SpeechTokenizer.

Table 9: Compression performance for latent codes of LLM-based entropy model on FreqCodec. "RVQ-1:32" denotes the sum from first layer to 32nd layer.

| | CR(%) | | |
|---|---|---|---|
| Settings | LibriTTS-clean | LibriTTS-other | LJSpeech |
| RVQ-1 | 92.19 | 97.45 | 91.44 |
| RVQ-1:2 | 95.60 | 98.73 | 95.22 |
| RVQ-1:8 | 98.90 | 99.68 | 98.81 |
| RVQ-1:32 | 99.73 | 99.92 | 99.70 |

As shown in Table. 9, the results demonstrate that the coding gain brought by the LLM-based entropy model to FreqCodec is much smaller than that for SpeechTokenizer. This is because the acoustic and semantic information are mixed in the RVQ indices obtained in each layer of FreqCodec. The gain of the LLM-based entropy model primarily benefits latent codes that contain semantic information. This is reflected in the satisfactory performance of our proposed entropy model on RVQ indices from the first layer of SpeechTokenizer. It guides the output RVQ indices from the first layer to contain semantic information through semantic distillation, separating it from the acoustic information.

