# OpenReview forum: "Large Language Models Are Stronger Entropy Models for Transform Coding"
_ICLR.cc/2025/Conference — Submitted to ICLR 2025_

### Official Review · Reviewer_WQ3m · 2024-10-28

**Soundness:** 3
**Presentation:** 3
**Contribution:** 3
**Rating:** 6
**Confidence:** 4

**Summary:**

This paper proposes a novel image and audio compression method that leverages large language models (LLMs) as entropy encoders, significantly enhancing the efficiency of traditional compression approaches. Unlike entropy encoding methods that rely on static probability distributions, LLMs can dynamically predict the conditional probability of each symbol by utilizing contextual information, resulting in more efficient encoding. By applying minimal fine-tuning to the LLM model, this method successfully adapts to data across different modalities.

**Strengths:**

1. The authors conducted multimodal testing of the method, covering various data types including images and audio, which validated the method's effectiveness across multiple data modalities. Experiments included a range of encoding baselines, such as JPEG, VQGAN, and SpeechTokenizer, demonstrating the method’s versatility, showing that LLMs can adapt to diverse data modalities and compression requirements.

2. Through dynamic probability prediction, the use of LLMs as entropy encoders significantly enhances compression efficiency, achieving notable improvements over baseline methods, particularly for image and audio data. Experimental results demonstrate that the model achieves strong compression performance across various datasets.

3. The paper presents a computationally resource-efficient approach that requires fine-tuning only a minimal portion of the LLM’s input and output layers to adapt to inputs of different dimensions.

**Weaknesses:**

1. This paper should provide experiments applying the LLM-based entropy encoding scheme to VAE-based models, such as HyperPrior, and conduct a comparative analysis with the results of the original entropy encoding scheme. This would yield more comprehensive results.

2. Table 4 is missing the inference time for the original VQGAN.

**Questions:**

The metrics section of the paper mentions the use of FID; however, it seems that this metric is not included in the comparative analysis of the experimental results.

---

> ### Author Response · Authors · 2024-11-25
> **Response to weaknesses and questions**
>
> Thank you very much for appreciating our contributions. We appreciate the opportunity to address the limitations and questions you mentioned:
>
>
> **Weakness-1:**"This paper should provide experiments applying the LLM-based entropy encoding scheme to VAE-based models, such as HyperPrior, and conduct a comparative analysis with the results of the original entropy encoding scheme. This would yield more comprehensive results."
>
> **Response-W1:**
>
> Thank you very much for your valuable suggestion. We will further explore the applications of our proposed coding scheme to VAE-based codecs and conduct comparative analysis in the following works.
>
>
> **Weakness-2:**"Table 4 is missing the inference time for the original VQGAN."
>
> **Response-W2:**
>
> We are sorry for the misunderstanding here. Table 4 in the paper refers to the running time of entropy coding using our proposed LLM-based entropy model. The following table shows the whole coding time of different anchors with our method for a 256 $\times$ 256 image, chunk size = 2048:
>
> |                   |         A        |            B            |         A+B         |            C            |             D            |          C+D          |
> |:-----------------:|:----------------:|:-----------------------:|:-------------------:|:-----------------------:|:------------------------:|:---------------------:|
> |       Anchor      | Transform Time/s | Entropy Encoding Time/s | Total Encoding Time | Entropy Decoding Time/s | Inverse-Transform Time/s | Total Decoding Time/s |
> |   JPEG vocab-256  |       0.013      |          1.718          |        1.731        |          1.712          |           0.007          |         1.719         |
> |  VQGAN vocab-1024 |       0.437      |          0.137          |        0.574        |          0.131          |           0.421          |         0.552         |
> | VQGAN vocab-16384 |       0.437      |          0.803          |        1.240        |          0.797          |           0.421          |         1.218         |
>
>
> **Question-1:**"The metrics section of the paper mentions the use of FID; however, it seems that this metric is not included in the comparative analysis of the experimental results."
>
> **Response-Q1:**
>
> Thank you for highlighting this. FID is utilized to assess the perceptual quality of VQGAN. The RD-Curve of VQGAN anchor using our proposed method is shown in **Figure 12** at **page 15** of our latest pdf.

---

> > ### Comment · Reviewer_WQ3m · 2024-11-27
> >
> > Thank you for your response. Some of my concerns have been addressed.

---

### Official Review · Reviewer_fGRF · 2024-11-03

**Soundness:** 3
**Presentation:** 3
**Contribution:** 2
**Rating:** 5
**Confidence:** 4

**Summary:**

In this paper, the authors proposed to use LLM + transform coding for the compression tasks. Unlike previous work which use LLM for performing compression in the raw signal (e.g. pixel) domain, the authods proposed to use LLM for performing compression in the latent domain, where the latent codes are obtained by transform like DCT or neural networks. The authors have demonstrated by simply finetuning LLM, it can be used for enhancing the compression performance of the latent code.

**Strengths:**

- In this paper, the authors propose using LLM + transform coding for compression tasks. I think this is a novel idea and worth studying.
- The authors demonstrated that, with simple fine-tuning, LLMs can be adapted for compressing the latent code generated by different transformations.

**Weaknesses:**

- In my opinion, the main contribution of the work is exploring the use of LLMs for compression in the latent domain. However, the techniques proposed for adapting LLMs for compressing different signals are straightforward.

- From this work, I can't see the potential for integrating LLMs into codecs (which the authors suggested) due to the model's size, complexity, and performance. The performance of the proposed method is clearly not comparable with SOTA codecs, requiring a large model with a significant amount of computation/parameters and a slow inference speed.
    - For example, it’s not comparable with the classical Hyperprior model in learned image compression, where the SOTA image codecs are far better than the Hyperprior model.
    - In the case with VQGAN, the number of required fine-tuning parameters is up to 134.2M, which is larger than most existing codecs. When the fine-tuning parameters exceed those of a neural codec (which also outperforms the proposed method), I can't agree that fine-tuning an LLM for compression is a practical approach.
    - While the use of an LLM within a codec for entropy coding may not be practical, I believe the study of LLMs for latent domain compression is still interesting.
- Although the authors suggested that the raster order preserves spatial correlations, I think this is not correct for the JPEG case, where the DCT converts the inputs into the frequency domain.
- As VQGAN was not originally proposed for the compression task, it may not be suitable as an anchor for evaluating compression performance.

**Questions:**

- Why is there no comparison to VQGAN in the RD plots?
- For the VQGAN anchor, how is entropy coding being performed?
- How are the rate points generated for Figure 5? (The authors only mentioned using QF 20/50/80, but there are more rate points in Figure 5.)
- Are only two rate points being used to calculate BD-rate for VQ-GAN+LLM?

---

> ### Author Response · Authors · 2024-11-25
> **Response to weaknesses**
>
> Thank you very much for recognizing our contributions. We greatly appreciate the opportunity to address the limitations and questions you have raised:
>
> **Weakness-1:**"In my opinion, the main contribution of the work is exploring the use of LLMs for compression in the latent domain. However, the techniques proposed for adapting LLMs for compressing different signals are straightforward."
>
> **Response-W1:**
>
> Thank you for your comment. Our simple yet effective method makes significant contributions as the first work to introduce an LLM-based entropy model for transform coding:
>
> - Make considerable BD-rate improvements on three typical kinds of codecs for compression.
> - Our proposed method is universal to dual-modal data including Speech and Image shown in this paper.
>
> **Weakness-2.1:**"For example, it’s not comparable with the classical Hyperprior model in learned image compression, where the SOTA image codecs are far better than the Hyperprior model."
>
> **Response-W2.1:**
>
> Thank you for your valuable suggestion. We will further explore the applications of our proposed coding scheme to VAE-based codecs and conduct comparative analysis in the following works.
>
>
> **Weakness-2.2:**"In the case with VQGAN, the number of required fine-tuning parameters is up to 134.2M, which is larger than most existing codecs. When the fine-tuning parameters exceed those of a neural codec (which also outperforms the proposed method), I can't agree that fine-tuning an LLM for compression is a practical approach."
>
> **Response-W2.2:**
>
> Thank you for your comment. The VQGAN was originally designed for generation rather than compression. A significant portion of 134.2M parameters stems from the large vocabulary size of its VQ codebook (16,384 in this case). In contrast, most compression methods typically utilize a much smaller vocabulary size for quantization. This means that for the vast majority of image and speech codecs, the parameters fine-tuned by our method will not exceed 33.6 MB.
>
>  | Vocab |    Applied Codec    | Params  |
>  |:-----:|:-------------------:|:-------:|
>  | 16384 | VQGAN (vocab-16384) | 134.2MB |
>  | 1024  | VQGAN (vocab-1024)  |  8.4MB  |
>  |  256  |        JPEG         |  2.1MB  |
>  | 4096  |   SpeechTokenizer   | 33.6MB  |
>
>
> **Weakness-2.3:**"While the use of an LLM within a codec for entropy coding may not be practical, I believe the study of LLMs for latent domain compression is still interesting."
>
> **Response-W2.3:**
>
> Thank you for recognizing our contributions. To the best of our knowledge, this is the first attempt at latent domain compression using LLM-based entropy model in this field, pushing coding efficiency to new heights for various kinds of codecs.
>
> ###
> **Weakness-3:**"Although the authors suggested that the raster order preserves spatial correlations, I think this is not correct for the JPEG case, where the DCT converts the inputs into the frequency domain."
>
> **Response-W3:**
>
> We observe that DCT coeffients still have strong 2D spatially-structural correlations and visualize them in **Figure 9** at **page 10** in the latest pdf. Specifically, each coeff. is correlated to neighboring coeffients.
>
>
> **Weakness-4:**"As VQGAN was not originally proposed for the compression task, it may not be suitable as an anchor for evaluating compression performance."
>
> **Response-W4:**
>
> Thank you for pointing this out. In our work, we choose VQGAN as one of the representative anchors for evaluations on both generation and compression.

---

> ### Author Response · Authors · 2024-11-25
> **Response to questions**
>
> **Question-1:**"Why is there no comparison to VQGAN in the RD plots?"
>
> **Response-Q1:**
>
> Thank you very much for reminding. We have added the RD-plot under original VQGAN codec and VQGAN codec with our proposed LLM-based entropy model in **Figure 12** at **Appendix 1.2.** (**page 15** of the latest pdf.)
>
> However, due to the limit of VQGAN codec itself, the VQGAN-based compression still CANNOT achieve SOTA performance in objective metrics like PSNR.
>
>
> **Question-2:**"For the VQGAN anchor, how is entropy coding being performed?"
>
> **Response-Q2:**
>
> Thanks for your question.
>
> VQGAN original procedure:
>
> - Stage I (VQ-VAE): Raw Images --> VQ Indices --> Recon. Images
> - Stage II (Transformer): VQ Indices --> Gen. Images
>
> VQGAN-based coding procedure:
>
> - Stage I: Unchanged as above.
> - Stage II: Encoding and Decoding.
>     - Encoding: Raw Images --> VQ Indices --> LLM's Probs. --> bitstream
>     - Decoding: bitstream + LLM's Probs. --> VQ Indices --> Recon. Images.
>
> The detailed procedure of VQGAN coding is described in **Figure 3 (b)** of the paper.
>
>
> **Question-3:**"How are the rate points generated for Figure 5? (The authors only mentioned using QF 20/50/80, but there are more rate points in Figure 5.)"
>
> **Response-Q3:**
>
> The nine points in Figure 5 are genertated by setting the quality factor of JPEG as [10, 15, 20, 40, 50, 80, 85, 90, 95]. We choose QF 20/50/80 of them to show the compression performance of our method on latent codes with different quality levels.
>
>
> **Question-4:**"Are only two rate points being used to calculate BD-rate for VQ-GAN+LLM?"
>
> **Response-Q4:**
>
> Yes, two rate points of VQGAN are from two pretrained models [1] with vocab sizes of 1024/16384.
>
> **References:**
>
>  [1] Esser, Patrick, Robin Rombach, and Bjorn Ommer. "Taming transformers for high-resolution image synthesis." Proceedings of the IEEE/CVF conference on computer vision and pattern recognition. 2021.

---

> ### Comment · Reviewer_fGRF · 2024-11-26
> **Further questions**
>
> Thank you to the authors for the response. I have the following further questions:
>
>
> DCT coefficients:
> - While I agree that there are correlations between the neighboring coefficients, I believe it is not the right term to call them "spatial correlations" as the coefficients represent different frequencies.
>
>
> VQGAN as an anchor:
> - I think the authors have not responded to the question regarding why VQGAN is being used for evaluating the compression performance.
>
>
> VQGAN anchor + entropy coding:
> - In that case, I believe that no entropy coding has been performed for the anchor? In that case, comparing the performance with the VQGAN baseline (which is neither designed for compression nor employs entropy coding) does not appear to be meaningful.
>
>
> In addition, I have noticed some other issues:
> - The citation for PSNR seems incorrect (the cited paper is not about the standard PSNR, which I believe is the one used in the paper?).
> - The reported runtime for JPEG appears unreasonably slow (3 seconds for 64x64, which is significantly slower than the case with VQGAN).

---

> ### Author Response · Authors · 2024-11-26
> **Response to further questions**
>
> We sincerely appreciate the timely response by the reviewer.
>
>
> **Question-1:** "While I agree that there are correlations between the neighboring coefficients, I believe it is not the right term to call them "spatial correlations" as the coefficients represent different frequencies."
>
> **Response-1:**
>
> Thanks for your agreement regarding the correlations between neighboring coefficients.
>
> Originally we would like to explain this "correlations" are related to neighboring similarity (i.e. "spatial correlations") in the pixel domain. Now we have corrected it to "correlations of different frequencies" in the latest pdf. Thanks for pointing it out.
>
>
> **Question-2:** "I think the authors have not responded to the question regarding why VQGAN is being used for evaluating the compression performance."
>
> **Response-2:**
>
> Thanks for the further question.
>
> While the VQGAN is originally designed for image generation, a number of recent works [1-4] have explored the potential of VQGAN as an anchor in image compression.
>
> The results of these studies demonstrate the significant potential of VQGAN framework in unifying image generation and compression. Therefore, our work adopts VQGAN as one of the representative anchors.
>
> **References:**
>
> [1] Mao, Qi, et al. "Extreme image compression using fine-tuned vqgans." Data Compression Conference (DCC) 2024.
>
> [2] Jia, Zhaoyang, et al. "Generative Latent Coding for Ultra-Low Bitrate Image Compression." CVPR 2024.
>
> [3] Xue, Naifu, et al. "Unifying Generation and Compression: Ultra-low bitrate Image Coding Via Multi-stage Transformer." arXiv preprint arXiv:2403.03736 (2024).
>
> [4] Li, Anqi, et al. "Once-for-All: Controllable Generative Image Compression with Dynamic Granularity Adaption." arXiv preprint arXiv:2406.00758 (2024).
>
>
> **Question-3:** "In that case, I believe that no entropy coding has been performed for the anchor? In that case, comparing the performance with the VQGAN baseline (which is neither designed for compression nor employs entropy coding) does not appear to be meaningful."
>
> **Response-3:**
>
> Sorry for the unspecific description of VQGAN-based coding procedure in the former response.
>
> VQGAN-based coding procedure in our work:
>
> - Encoding: Raw Images --> VQ Indices --> LLM's Probs. --> **Arithmetic Encoder** --> bitstream
>
> - Decoding: bitstream + LLM's Probs. --> **Arithmetic Decoder** --> VQ Indices --> Recon. Images.
>
> In conclusion, our proposed LLM-based entropy model further compress the VQ indices, regarding VQGAN as a anchor.
>
>
> **Question-4:** "The citation for PSNR seems incorrect (the cited paper is not about the standard PSNR, which I believe is the one used in the paper?)."
>
> **Response-4:**
>
> Thanks a lot for pointing this out. The PSNR used in this paper is the standard one. We have corrected the improper citation in the updated pdf.
>
>
> **Question-5:** The reported runtime for JPEG appears unreasonably slow (3 seconds for 64x64, which is significantly slower than the case with VQGAN).
>
> **Response-5:**
>
> Thank you for your question.
>
> The reported runtime refers to the sum of entropy encoding and decoding time, using our autoregressive method on different anchors. The runtime is then proportional to the size of latent codes.
>
> - **For JPEG anchor**, the size of DCT latent codes is the same as the image size. That means **4,096 codes** to be autoregrassively compressed for a 64 $\times$ 64 image. The total running time is equal to 4096 multiplied by one-token LLM inference time.
>
> - **For VQGAN anchor**, the size of latent codes downsampled by encoder is much smaller than the original image size. That means **256 codes** (16 $\times$ 16) to be autoregrassively compressed for a 256 $\times$ 256 image. The total running time is equal to 256 multiplied by one-token LLM inference time.
>
> - That's why the reported runtime for JPEG (4,096 codes) is much slower than that of VQGAN (256 codes). Actually, recent neural codecs generate latent codes with a downsampled size relative to the image size. **By applying the LLM-based entropy model to latent codes (compact) instead of raw data (redundant), the running time of the autoregressive LLM-based entropy model is reduced to a more practical level.** This is one of benefits and motivations of our work.

---

> ### Comment · Reviewer_fGRF · 2024-11-26
>
> Thank you the authors for providing further clarifications.
>
> Regarding Question 3, to be more specific, I was asking whether the baseline VQGAN pipeline (without the addition of the LLM) incorporates any entropy coding. If no entropy coding is utilized in the VQGAN (without LLM), the reported 17% improvement may not represent a significant gain, given that VQGAN itself is not designed for compression tasks.

---

> > ### Author Response · Authors · 2024-11-27
> >
> > Thanks a lot for the very patient explanation of Question 3 by the reviewer.
> >
> > VQ indices are not usually subjected to further compression. That's why we did not include an entropy coder in the baseline of VQGAN. However, the insightful question by you the reviewer piques our interest.
> >
> > We analyze the probability distribution of the VQ indices and generate the Huffman probability table, on the Kodak dataset. Subsequently, we perform compression using Huffman coding based on this analysis.
> >
> > The table below compares the compression rates of Huffman coding and our method on VQ indices, evaluated on the Kodak dataset.
> >
> > |  Entropy Model | CR(%) at VQGAN (vocab-1024) | CR(%) at VQGAN (vocab-16384) |
> > |:--------------:|:---------------------------:|:----------------------------:|
> > | Huffman Coding |            96.31            |        85.05                 |
> > |      Ours      |            90.70             |                75.41        |
> >
> > The results demonstrate that our LLM-based entropy model is a stronger entropy model, achieving better compression performance on VQ indices with varying vocab. sizes.

---

> > > ### Comment · Reviewer_fGRF · 2024-12-03
> > >
> > > Thank you to the authors for the further reply.
> > >
> > > In this case, the improvement achieved with the use of the LLM is only ~6% and ~11% compared to the unconditional distribution with Huffman coding for the vocab-1024 and vocab-16384 cases, respectively. Considering the use of context modeling, the large number of fine-tuning parameters and computational resources required, I cannot agree that the proposed method has great potential for improving coding efficiency, as claimed in the paper.

---

### Official Review · Reviewer_cjvn · 2024-11-03

**Soundness:** 2
**Presentation:** 2
**Contribution:** 2
**Rating:** 5
**Confidence:** 4

**Summary:**

The paper proposes a method to leverage the strength of LLMs for predictive coding in the transform-coding setting. Specifically, the paper proposes a latent code arrangement technique that is shown to work with three different codecs (JPEG, VQGAN, and SpeechTokenizer). The latent code arrangement is used to align the dimensions of the LLMs and the latent codes. The input and output layers of the LLM are trained while the rest of the LLM backbone is frozen. The proposed approach delivers significant improvements with JPEG and consistent improvements with VQGAN and SpeechTokenizer.

**Strengths:**

1. The proposed approach delivers impressive gains with JPEG and consistent gains with VQGAN and SpeechTokenizer.

2. The latent code arrangement approach is flexible, which makes it well-suited to the transform-coding framework.

**Weaknesses:**

1. Lack of novelty. The key idea of leveraging the strong predictive capabilities of LLMs for compression has been proposed and explored in the literature (Deletang et al. 2024, Valmeekam et al. 2023). Given this, the main contribution in this work (of latent code arrangement) is incremental as far as novelty is concerned.

2. There is no qualitative evaluation of the proposed approach. It would be interesting to see how the coding efficiency manifests in terms of perceptual image/speech quality.

3. There is no comparison with Deletang et al.’s work. Given the same underlying philosophy, such a comparison is warranted.

4. There is no discussion on the choice and size of the training dataset on performance. Also, the ImageNet dataset is not particularly suited for evaluating codec performance.

5. The quality of the writing needs improvement. For example, there are missing references and grammatical errors.

**Questions:**

1. What is the impact of the coding gains on perceptual quality? Can you please provide image examples?

2. Have you considered other perceptual quality metrics for evaluating performance?

3, Why is a comparison with related methods not presented? Such a comparison would help place the proposed work on a strong footing.

4. What is the impact of the training dataset on performance? What is the impact of using the Kodak dataset for testing instead of ImageNet?

5. Why was the ImageNet dataset used for training the model? ImageNet data is not particularly suited for testing codec performance. Questions 4 and 5 apply to the speech codec as well.

6. What is the impact of the inference time on scaling the proposed method to large data?

7. Why are smaller image sizes considered for running time measurements? These sizes are not representative of standard image resolutions such as full-HD or higher.

---

> ### Author Response · Authors · 2024-11-25
> **Response to weaknesses**
>
> Thank you very much for appreciating our contributions. We appreciate the opportunity to address the limitations and questions you mentioned:
>
>
> **Weakness-1:**"Lack of novelty. The key idea of leveraging the strong predictive capabilities of LLMs for compression has been proposed and explored in the literature (Deletang et al. 2024, Valmeekam et al. 2023). Given this, the main contribution in this work (of latent code arrangement) is incremental as far as novelty is concerned.”
>
> **Response-W1:**
>
> Thanks for your comment. In fact, it is widely recognized that transformed coefficients (such as those obtained through Fourier Transforms [1], Wavelet Transforms [2], nonlinear neural transforms [3], or STFT [4]) are generally easier to compress, particularly for images and speeches. This understanding motivates us to explore LLM-based compression applied to transformed coefficients, a direction that has yet to be investigated.
>
> Our simple yet effective method makes significant contributions as the first work to introduce an LLM-based entropy model for transform coding:
>
> - Make considerable BD-rate improvements on three typical kinds of codecs for compression.
> - Our proposed method is universal to dual-modal data including Speech and Image shown in this paper.
>
> We personally hope this work would provide comprehensive experimental evidence on this promising topic.
>
> **References:**
>
> [1] Pratt, W.K., Kane, J., Andrews, H.C. "Hadamard transform image coding". Proceedings of the IEEE. 57:58–68. 1969.
>
> [2] Unser, M., Blu, T. "Mathematical properties of the JPEG2000 wavelet filters", IEEE Transactions on Image Processing. 12 (9): 1080–1090, 2023.
>
> [3] Johannes Ballé et al. "Nonlinear Transform Coding", IEEE Journal of Selected Topics in Signal Processing 2020.
>
> [4] Zhihao Du and Shiliang Zhang and Kai Hu and Siqi Zheng, "FunCodec: A Fundamental, Reproducible and Integrable Open-source Toolkit for Neural Speech Codec", arXiv preprint 2309.07405.
>
>
> **Weakness-2**"There is no qualitative evaluation of the proposed approach. It would be interesting to see how the coding efficiency manifests in terms of perceptual image/speech quality."
>
> **Response-W2:**
>
> Thank you for your question. The compression procedure of our LLM-based entropy model on the latent codes is **lossless** (quote **section 2** in the paper). That's why we emphasize "Stronger Entropy Model" in our title. Therefore, the perceptual quality of reconstructed images and speches using our method is just as the same as that using original anchor codecs.
>
>
> **Weakness-3:**"There is no comparison with Deletang et al.’s work. Given the same underlying philosophy, such a comparison is warranted."
>
> **Respond-W3:**
>
> Thank you for your comment. Firstly, we choose Llama3-8B as the backbone of our LLM-based entropy model, based on the comparison with Deletang's work on lossless text compression task as mentioned in **Table 1** of the paper.
>
> Secondly, Deletang et al's work is proposed for compressing data in raw domain and **CANNOT** be directly applied in latent domain.
>
>
> **Weakness-4:**"There is no discussion on the choice and size of the training dataset on performance. Also, the ImageNet dataset is not particularly suited for evaluating codec performance."
>
> **Response-W4:**
>
> Thank you for reminding. We have added the discussion on the choice and size of the training dataset in **section 5.1** of the latest paper.
>
> We acknowledge that the ImageNet dataset is not a typical choice for evaluating codec performance. We include results on the ImageNet validation subset to demonstrate the robustness of our method in cross-data validation.
>
>
> **Weakness-5:**"The quality of the writing needs improvement. For example, there are missing references and grammatical errors."
>
> **Response-W5:**
>
> Thank you very much for reminding. We have addressed the issues you mentioned and further improved our writing in our latest paper.

---

> ### Author Response · Authors · 2024-11-25
> **Response to questions**
>
> **Question-1:**"What is the impact of the coding gains on perceptual quality? Can you please provide image examples?"
>
> **Response-Q1：**
>
> As mention in the **Response** to **Weakness-2**. Our proposed LLM-based entropy model compress the latent codes losslessly. That is, our method can get the same perceptual quality of reconstructed data as anchor codecs, but with less bitrate.
>
>
> **Question-2:**"Have you considered other perceptual quality metrics for evaluating performance?"
>
> **Response-Q2:**
>
> Thank you for your suggestion. We have added MS-SSIM to evaluate compression performance of image codecs, as shown in **Figure 11** of the latest pdf.
>
>
> **Question-3:**"Why is a comparison with related methods not presented? Such a comparison would help place the proposed work on a strong footing."
>
> **Response-Q3:**
>
> Sorry for causing your misunderstanding on our contribution.
>
> Our work is **lossless** compression.  To our knowledge, we are the first work to introduce LLM-based entropy model for transform coding. As mentioned in **Fig 5** of the paper, "Ours+JPEG" exceeds JPEG2000, WebP, FactorizedPrior and catches up with HyperPrior and BPG.
>
>
> **Question-4:**"What is the impact of the training dataset on performance? What is the impact of using the Kodak dataset for testing instead of ImageNet?"
>
> **Response-Q4:**
>
> Thank you for your questions. We test the compression performance on ImageNet validation subset to ensure the robustness of our method in cross-data validation.
>
> The Kodak dataset is commonly used to evaluate compression performance of image codecs with 24 $\times$ high-quality images.
>
>
> **Question-5:**"Why was the ImageNet dataset used for training the model? ImageNet data is not particularly suited for testing codec performance. Questions 4 and 5 apply to the speech codec as well."
>
> **Response-Q5:**
>
> Thank you for your comment. ImageNet is a widely-used dataset for training in learnable image compression. As for the speech codecs, we follow the training method in [5].
>
> **References:**
>
> [5]Zhang, Xin, et al. "Speechtokenizer: Unified speech tokenizer for speech large language models." arXiv preprint arXiv:2308.16692 (2023).
>
>
> **Question-6:**"What is the impact of the inference time on scaling the proposed method to large data?"
>
> **Response-Q6:**
>
> Thank you for your question.
> The running time of image anchor with the chunk size of 2048 on different image sizes is shown in the following table:
>
> |          Running Time(s)          | 64x64 | 256x256 | 768x512 | 1920x1080(full-HD) |
> |:---------------------------------:|:-----:|:-------:|:-------:|:------------------:|
> |     Ours+JPEG     | 3.43  |  54.88  | 329.28  |      1736.44       |
> | Ours+VQGAN (vocab-1024)  | 0.017 |  0.27   |  1.60   |        8.45        |
> | Ours+VQGAN  (vocab-16384)  | 0.10  |  1.60   |  9.60   |       50.63        |
>
> Without parallel computation, the inference time of our autoregressive method is proportionally increasing when scaling to large data.
>
>
> **Question-7:** "Why are smaller image sizes considered for running time measurements? These sizes are not representative of standard image resolutions such as full-HD or higher."
>
> **Response-Q7:**
>
> Thank you for pointing this out. Regarding the image sizes, the running time of image anchor with the chunk size of 2048 on different sizes is shown in the following table:
>
> |          Running Time(s)          | 64x64 | 256x256 | 768x512 | 1920x1080(full-HD) |
> |:---------------------------------:|:-----:|:-------:|:-------:|:------------------:|
> |     Ours+JPEG     | 3.43  |  54.88  | 329.28  |      1736.44       |
> | Ours+VQGAN (vocab-1024)  | 0.017 |  0.27   |  1.60   |        8.45        |
> | Ours+VQGAN (vocab-16384) | 0.10  |  1.60   |  9.60   |       50.63        |
>
>
> In fact, we have not yet conducted any parallel computation, and LLMs are autoregressivly computed. Then the running time is directly proportional to the image size. For JPEG, we just choose the size of 64x64 as a typical size of coding unit in our paper. For other neural codecs, we evaluate them using size of 256x256.

---

### Official Review · Reviewer_TGwR · 2024-11-04

**Soundness:** 3
**Presentation:** 2
**Contribution:** 2
**Rating:** 3
**Confidence:** 5

**Summary:**

This paper studies using LLMs in the compression of transformed data in lossy compression. It is not about using LLMs to improve the transformations, but only about improving probability estimation for the arithmetic coding stages. It uses the power and computational complexity of LLM to generate better probabilities for transform data, which are coded using arithmetic coding.

In the paper most of the work is about ad hoc modifications to adapt the data to format most suitable for LLMs, and related training and adjustments.

**Strengths:**

The idea of using LLMs for estimating symbol probabilities in entropy coding is not new. In fact, since it represents a more complex prediction structure, it is not surprising that it could be used in applications like lossless compression of text and somewhat improve results. There are several papers on the subject (including in the citations of this paper).

The main contribution of this paper is to extend this approach to compressing data after linear or nonlinear transformations. Since the purpose of those transformations is exactly to reduce source redundancies, they by definition are meant to use simpler entropy models. Thus, compression results are better, but it is not surprising that a very complex method will enable better compression.

**Weaknesses:**

While the results do so show some improvement, they are mostly when compared to much simpler methods. For example, for image coding the results may seem significant when compared to the 40+-year-old coding techniques of JPEG, but much less when compared to the hyper-prior prediction, which is much simpler neural-based approach and not sufficiently discussed.

It is important to consider that the most fundamental practical problem is if the improvement in compression justifies such very expensive LLM computations. Comparisons with methods that are orders of magnitude more efficient are not fair, and do not provide real insight on the contribution.

Complexity is partially shown by running times in Table 4, but this table is not even commented anywhere in the text. The caption refers to 64x64 images, but that size is completely unrealistic. This needs to be more carefully analyzed and explained.

In summary, paper claims to be the first to apply LLMs to help entropy coding of transformed data, and as expected it gives some improvement due to the better prediction capabilities of LLMs, but with enormously larger complexity, and the results are not significantly better than other simpler neural-network-based prediction like hyperprior networks, or entropy coding methods more modern and more powerful than JPEG’s.

From a theoretical perspective, it is not clear if there is enough novelty in showing that LLM-based compression, which has been shown to work on other data, also works with transform coefficients.

**Questions:**

It would be interesting to see more comparisons with the hyper-prior approaches (including complex recursive version), and computational complexity.

---

> ### Author Response · Authors · 2024-11-25
> **Response to weaknesses**
>
> Sincere thanks for your valuable suggestions. We greatly appreciate the opportunity to address the weaknesses:
>
> **Weakness-1 :**"While the results do so show some improvement, they are mostly when compared to much simpler methods."
>
> **Response-W1:**
>
> Thank you for your insightful comment. Our straightforward yet effective approach represents the first empirical analysis showcasing the significant potential of the LLM-based entropy model for achieving further compression across various codecs.
>
> We hope this work serves as a foundation by providing comprehensive experimental evidence on this promising topic. In future work, we plan to extend our proposed coding scheme to recent hyperprior-based methods and conduct comparative analyses to further validate its effectiveness.
>
>
> **Weakness-2:**"It is important to consider that the most fundamental practical problem is if the improvement in compression justifies such very expensive LLM computations. Comparisons with methods that are orders of magnitude more efficient are not fair, and do not provide real insight on the contribution."
>
> **Response-W2:**
>
> Thank you for your comment. The main contribution is to validate the effectiveness of our proposed idea to compress transform coefficients by leveraging LLMs, i.e. significant coding gains of 54.07% over JPEG, 17.61% over VQGAN and 34.61% over SpeechTokenizer. Meanwhile, we believe that future advancements in low-complexity LLMs will further enhance the practicality of this approach.
>
>
> **Weakness-3:**"Complexity is partially shown by running times in Table 4, but this table is not even commented anywhere in the text. The caption refers to 64x64 images, but that size is completely unrealistic. This needs to be more carefully analyzed and explained."
>
> **Response-W3:**
>
> Apologies for the missing reference to Table 4. We have included the relevant comments to **"Limitations in Compression Complexity"**, in Section 4 of the updated paper.
>
> Regarding image sizes, the running time of JPEG anchor with the chunk size of 2048 on different sizes is shown in the following table:
>
> |      Running Time(s)      |  8x8  | 64x64 | 256x256 |
> |:-------------------------:|:-----:|:------:|:-------:|
> | JPEG + Ours | 0.054 (1x) |  3.43 (~64x)  |  54.88 (~1024x)  |
>
> In fact, we have not yet conducted any parallel computation, and LLMs are autoregressivly computed. Then the running time is directly proportional to the image size. For JPEG, We just choose a typical coding unit size of 64x64 in our paper. For other neural codecs, we evaluate them using the training size of 256x256.
>
>
> **Weakness-4:**"In summary, paper claims to be the first to apply LLMs to help entropy coding of transformed data, and as expected it gives some improvement due to the better prediction capabilities of LLMs, but with enormously larger complexity, and the results are not significantly better than other simpler neural-network-based prediction like hyperprior networks, or entropy coding methods more modern and more powerful than JPEG’s."
>
> **Response-W4:**
>
> Thanks for your comment. To our knowledge, this is the first work to introduce an LLM-based entropy model for transform coding:
>
> - Make considerable BD-rate improvements on three typical kinds of codecs. e.g. Let JPEG (1992) catch up with HyperPrior (2018).
> - Our method is universal to dual-modal data including Speech and Image, shown in this paper.
>
> We personally hope this work would provide comprehensive experimental evidence on this promising topic. And we will conduct experiments on hyperprior networks in the future.
>
>
> **Weakness-5:**"From a theoretical perspective, it is not clear if there is enough novelty in showing that LLM-based compression, which has been shown to work on other data, also works with transform coefficients."
>
> **Response-W5:**
>
> Thanks for your comment. It is generally believed transformed coefficients (such as Fourier Transforms[1], Wavelet Transforms[2], nonlinear neural transforms[3] or STFT[4]) are easier to compress, especially for images and speeches. Then it motivates us to conduct LLM-based compression to transform coefficients, instead of raw data.
>
> Then, we select JPEG (DCT), VQGAN (neural transforms), SpeechTokenizer (STFT) as our anchors, and this has not been investigated before. By experiments, we provide evidence on this promising topic. However, the interpretability of LLMs remains an open challenge, and we plan to continue exploring this aspect further.
>
> **References:**
>
> [1] Pratt, W.K., et al. "Hadamard transform image coding". Proceedings of the IEEE. 57:58–68. 1969.
>
> [2] Unser, M., Blu, T. "Mathematical properties of the JPEG2000 wavelet filters", IEEE Transactions on Image Processing. 12 (9): 1080–1090, 2023.
>
> [3] Ballé et al. "Nonlinear Transform Coding", IEEE Journal of Selected Topics in Signal Processing 2020.
>
> [4] Zhihao Du et al., "FunCodec: A Fundamental, Reproducible and Integrable Open-source Toolkit for Neural Speech Codec", arXiv preprint 2309.07405.

---

> ### Author Response · Authors · 2024-11-25
> **Response to questions**
>
> **Question-1：**"It would be interesting to see more comparisons with the hyper-prior approaches (including complex recursive version), and the computational complexity."
>
> **Respond-Q1:**
>
> Thank you very much for the precious suggestion.
>
> **Regarding to Comprisons with Hyperprior Methods:**
>
> We will further explore the applications of our proposed coding scheme to VAE-based codecs and conduct comparative analysis in the following works.
>
>
> **Regarding to Computational Complexity:**
>
> We have added the reference of Table 4 at the discussion on **"Lmitations in Compression Complexity"** in **section 4** of the latest paper.
>
> Regarding the image sizes, the running time of image anchor with the chunk size of 2048 on different sizes is shown in the following table:
>
> |          Running Time(s)          | 64x64 | 256x256 | 768x512 | 1920x1080(full-HD) |
> |:---------------------------------:|:-----:|:-------:|:-------:|:------------------:|
> |     Ours+JPEG     | 3.43  |  54.88  | 329.28  |      1736.44       |
> | Ours+VQGAN (vocab-1024)  | 0.017 |  0.27   |  1.60   |        8.45        |
> | Ours+VQGAN (vocab-16384) | 0.10  |  1.60   |  9.60   |       50.63        |
>
> In fact, we have not yet conducted any parallel computation, and LLMs are autoregressivly computed. Then the running time is directly proportional to the image size. For JPEG, we just choose the size of 64x64 as a typical size of coding unit in our paper. For other neural codecs, we evaluate them using size of 256x256.

---

> > ### Comment · Reviewer_TGwR · 2024-11-26
> >
> > The authors replies are mostly about claiming to be the first to apply LLMs to entropy-code transform coefficients. However, since there are previous works showing the application to other entropy-coding problems, the novelty of just applying LLMs to this type of data is not significant and would need to be complemented with thorough analysis and comparisons.
> >
> > The important weakness of not comparing to other neural approaches, like hyper-prior networks, is recognized by the authors, but only to say that it will be in future works. This means the current version of the paper still has very important comparisons and analysis missing.

---

### Meta-Review · Area_Chair_nSnM · 2024-12-16

**Metareview:**

The paper discusses using the predictive capabilities of LLMs in the context of transform coding of images, in order to improve the entropy coding stage. This may represent the first study doing so.

However, as noted by the reviewers, important comparisons to recent work in the area of nonlinear transform coding (NTC) are missing, and the provided comparisons to the hyperprior model and to VQGAN with a factorized entropy model (Huffman coding) do not support the claim that LLMs are "stronger entropy models". It is entirely possible the demonstrated gains will evaporate when more appropriate comparisons are made. Hence, the current results do not justify the complexity of LLMs in the context of neural compression, where computational complexity is an important factor determining the practicality of a method.

Further work is needed to harness the predictive power of LLMs for this type of application.

**Additional Comments On Reviewer Discussion:**

Most reviewers pointed out that the empirical results are unconvincing. While the authors took a few steps to address the issues, they did not ultimately provide convincing evidence that using LLMs in this context is worth the complexity.

---

### Decision · Program_Chairs · 2025-01-22

Reject